# DeCoDEx: Confounder Detector Guidance for Improved Diffusion-based Counterfactual Explanations

**Nima Fathi**[*,1,2]                                           NIMAFH@CIM.MCGILL.CA

**Amar Kumar**[*,1,2]                                           AMARKR@CIM.MCGILL.CA

**Brennan Nichyporuk**[2]                                   BRENNANN@CIM.MCGILL.CA

**Mohammad Havaei**[3]                                       MHAVAEI@GOOGLE.COM

**Tal Arbel**[1,2]                                                   ARBEL@CIM.MCGILL.CA

[1] *Center for Intelligent Machines, McGill University, Montreal, Canada.*

[2] *MILA (Quebec AI institute), Montreal, Canada.*

[3] *Google Research, Montreal, Canada.*

**Editors:** Accepted for publication on at MIDL 2024

## Abstract

Deep learning classifiers are prone to latching onto dominant confounders present in a dataset rather than on the causal markers associated with the target class, leading to poor generalization and biased predictions. Although explainability via counterfactual image generation has been successful at exposing the problem, bias mitigation strategies that permit accurate explainability in the presence of dominant and diverse artifacts remain unsolved. In this work, we propose the DeCoDEx framework and show how an external, pre-trained binary artifact detector can be leveraged during inference to guide a diffusion-based counterfactual image generator towards accurate explainability. Experiments on the CheXpert dataset, using both synthetic artifacts and real visual artifacts (support devices), show that the proposed method successfully synthesizes the counterfactual images that change the causal pathology markers associated with Pleural Effusion while preserving or ignoring the visual artifacts. Augmentation of ERM and Group-DRO classifiers with the DeCoDEx generated images substantially improves the results across underrepresented groups that are out of distribution for each class. The code is made publicly available at https://github.com/NimaFathi/DeCoDEx.

**Keywords:** Bias Mitigation, Causality, Counterfactual Image Synthesis, Diffusion, Explainability, Spurious Correlations

## 1. Introduction

Deep learning (DL) methods have shown tremendous success in a wide variety of medical image tasks, including disease classification, due to their ability to learn generalizable, discriminative features across subjects. However, DL models are prone to learning shortcuts in order to obtain high overall accuracies, including any prevalent visual artifacts (e.g. marks in the image (DeGrave et al., 2021)) that are correlated with, but not causal of, the target outcome. Models that have not learned the relevant causal visual markers (Jia and Liang, 2017; Zech et al., 2018) are *right for the wrong reasons* (Sun et al., 2023b,a), and fail to

---

[*] Contributed equally

generalize across out-of-distribution subgroups (Geirhos et al., 2018, 2020). Explainable DL models that not only expose these biases but mitigate them are required in order to ensure their trustworthiness for safe clinical deployment.

Counterfactual (CF) image generation methods (e.g. Gifsplanation (Cohen et al., 2021) and Attri-Net (Sun et al., 2023b,a)) have recently been successful at exposing when the classifier is latching onto spurious correlations in order to obtain high performance. These methods employ conditional generation of the counterfactual image when the classifier has the opposing target outcome. Differences between the factual and counterfactual images should reflect the predictive local markers indicative of the class label, but also expose the classifier's reliance on spurious correlations. These methods do not mitigate the biases nor address their poor generalization. A number of debiasing methods (Sagawa et al., 2020; Wang et al., 2020; Sarhan et al., 2020) have recently been successful in several medical imaging contexts. Recent work (Kumar et al., 2023) combined Cycle-GAN counterfactual image generation and a Group-DRO (Sagawa et al., 2020) classifier to expose and mitigate the biases. The results showed improvement for minority subgroups, and classification based on disease-specific features. However, debiasing techniques have a number of known drawbacks, including improved fairness at the expense of a reduction in the performance in the majority subgroup. Integrating them into the counterfactual models requires sub-group labels for each class during training (which is often unavailable), and GANs require retraining the generative model with the pre-trained debiasing classifier in order to provide supervision for the counterfactual synthesis, which makes the process inflexible. Recently, an unconditional DDPM (Denoising Diffusion Probabilistic Model) (Ho et al., 2020), DiME (Jeanneret et al., 2022), has been proposed to generate classifier-guided counterfactual explanations, however, with the classifier latching onto shortcuts present in the dataset. Overall, developing bias mitigation strategies that permit accurate explainability in the presence of dominant and diverse visual artifacts remain open research questions.

This paper introduces DeCoDEx, a diffusion-based (DDPM) counterfactual image generator for debiased classifier explainability in the presence of dominant and diverse visual artifacts. DeCoDEx overcomes a number of limitations of current approaches: Rather than requiring training specialized debiasing classifiers for known subgroups, and then retraining the counterfactual generator that uses them (e.g. GANs), the framework provides debiased explainability of the classifier in question *at inference time* by leveraging the flexibility and explicit inference procedure (Dhariwal and Nichol, 2021; Wang et al., 2022; Kazerouni et al., 2022) of DDPMs, is generalizable to any subgroups, and shows stable training as compared to GANs. The framework can make use of any pre-trained, binary detector trained to indicate the presence or absence of the visual artifact in question (for any classes). During inference, the detector guides the diffusion-based counterfactual image generator towards accurate explainability, as the gradients from the detector counter the gradients from the classifier away from spurious correlations.

Extensive experiments are performed on the publicly available CheXpert dataset (Irvin et al., 2019), using both synthetic artifacts and real visual artifacts (support devices). Qualitative results show that the proposed method successfully synthesizes the counterfactual images by making changes in the pathology associated with Pleural Effusion while preserving or ignoring the visual artifacts. The quality of the counterfactual images was measured via several metrics, such as L1 distance, Counterfactual Prediction Gain (CPG) and Spuri-

ous Correlation Latching Score (Kumar et al., 2023). The results indicated the strength of framework over a baseline (without a detector). Augmentation of the dataset with counterfactual images synthesized with DeCoDEx improves ERM and Group-DRO classification for the minority subgroups.

## 2. Methodology

The DeCoDEX framework involves a training strategy for explainability via counterfactual image synthesis, while ensuring robustness to spurious correlations. The model consists of a classifier and a trained unconditional denoising diffusion probabilistic model (DDPM). The framework leverages a pre-trained visual artifact detector that guides a DDPM to synthesize counterfactuals while ignoring the artifact.

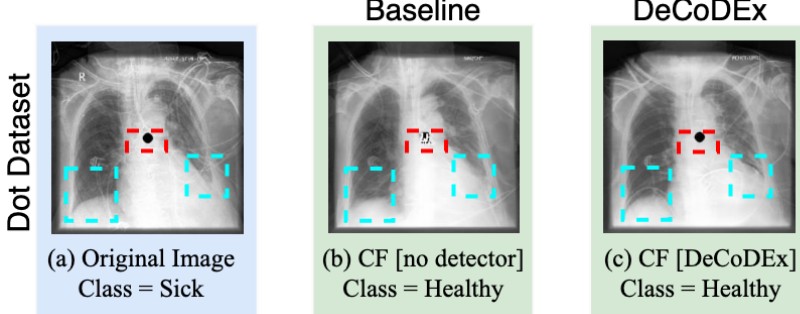

Figure 1: CF explanations for a subject with Pleural Effusion in the presence of an artifact: (a) Chest radiograph of a sick patient: dot artifact, disease pathology; (b) CF image from biased classifier using DDPM (i.e. DeCoDEx without detector) maintains the diseased area but modifies the dot; (c) DeCoDEx CF image modifies the Pleural Effusion area to look healthy as expected (Huang et al., 2022; Wang et al., 2017) while ignoring the dot artifact.

One of the advantages of our approach is the flexibility to use any pre-trained detector that can identify spurious correlations in the input sample. During the counterfactual image generation, when the biased classifier relies on spurious non-causal features in the dataset, the detector's gradient reversal signal readjusts the generation process, steering it back toward focusing on relevant pathological features (Fig. 1). It should be noted here that if there are several spurious correlations in the dataset that are difficult to detect, the detector may only block only some of them. Therefore, for some difficult cases, even after using a detector the counterfactual images may fail to make changes in the area correlated with the target class. An overview of our method is shown in Fig. 2.

**Counterfactual Image Generation**: The counterfactual image generation is designed to adhere to a set of constraints (Mothilal et al., 2020; Nemirovsky et al., 2020) through the following loss functions: (i)*Identity preservation loss*, $\mathcal{L}_{perc}$: counterfactual images preserves the identity of the factual image, $\mathcal{L}_{perc}$; (ii) *Classifier consistency loss*, $\mathcal{L}_{class}$: counterfactual image belongs to the correct target class. An additional *Detector loss* term, $\mathcal{L}_{det}$, is intro-

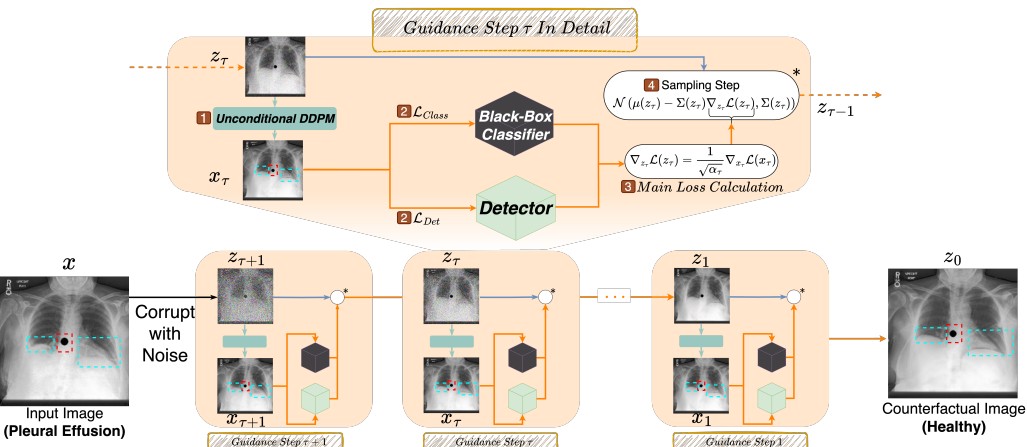

Figure 2: DeCoDEx Framework: Generating the counterfactuals (CFs) involves several inference steps. At each step, there are several components: (1) Denoising via unconditional DDPM, (2) pretrained classifier and detector loss, (3) gradient of the classifier, detector and perceptual loss, (4) counterfactual synthesis via sampling and backpropagating loss from black-box classifier and detector. The classifier, detector and unconditional DDPM are all pre-trained components. The resulting CF makes changes to the disease markers while disregarding visual artifacts.

duced to help guide the generation away from the spurious correlations that the classifier would have otherwise latched onto.

The generative model for synthesizing counterfactual images use a pre-trained DDPM with classifier guidance. DDPMs operate through a forward diffusion process (Eq. 1), which incrementally corrupts the original image $x$ (or $z_0$) by adding Gaussian noise, culminating in a highly noised image $z_T$ over $T$ timesteps.

$$z_t \sim \mathcal{N}\left(\sqrt{1-\beta_t}z_{t-1}, \beta_t \mathbb{I}\right), \tag{1}$$

where $\beta_t$ are predefined noise levels. The reverse diffusion process (Eq. 2) aims to recover the original image from $z_T$, using a neural network to predict and subtract the added noise iteratively.

$$x_{t-1} = \frac{1}{\sqrt{\alpha_t}}\left(x_t - \frac{1-\alpha_t}{\sqrt{1-\bar{\alpha}_t}}\epsilon_\theta(x_t, t)\right), \tag{2}$$

where $\alpha_t := \prod_{k=1}^{t}(1-\beta_k)$ and $\epsilon_\theta(x_t, t)$ is the noise estimated by the network at step $t$. Guided-diffusion sampling (Dhariwal and Nichol, 2021) is used to denoise the image at any time step $t$, given by $z_{t-1} \sim \mathcal{N}\left(\mu(z_t) - \Sigma(z_t)\nabla_{z_t}\mathcal{L}(z_t; y_c, y_s, x_t), \Sigma(z_t)\right)$. The complete loss function is given by:

$$\mathcal{L}(x_t; y_c, y_s, x) = \lambda_c \mathcal{L}_{class}\left(C(y_c|x_t)\right) + \lambda_d \mathcal{L}_{det}\left(D(y_s|x_t)\right) + \lambda_p \mathcal{L}_{perc}(x_t, x), \tag{3}$$

where $C$ and $D$ refer to our classifier and detector modules, $y_c$ and $y_s$ refer to the target labels of the class and the presence of spurious correlation and $\lambda_c, \lambda_d, \lambda_p$ are hyperparmeters.

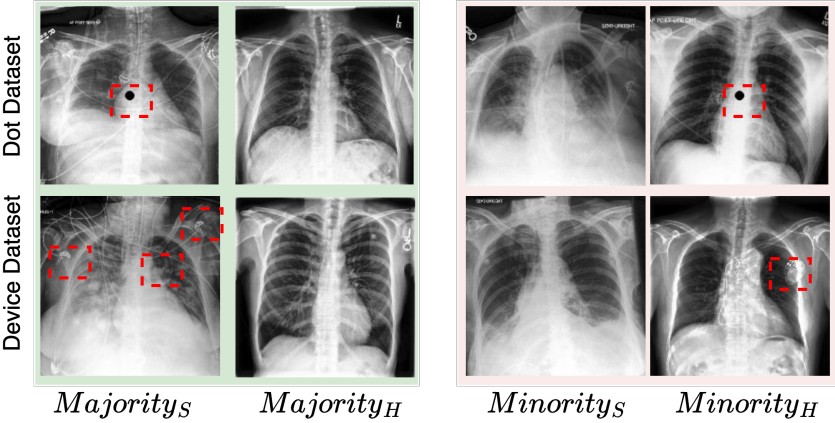

Figure 3: Majority and Minority subgroup samples from the Dot dataset (top row) and Device dataset (bottom row). Red boxes show the location of the artifacts.

Finally, the gradient of the complete loss function can be expressed as follows:

$$\nabla_{z_t}\mathcal{L}(z_t; y_c, y_s, x_t) = \frac{1}{\sqrt{\alpha_t}}\nabla_{x_t}\mathcal{L}(x_t; y_c, y_s, x). \tag{4}$$

## 3. Experiments and Results

### 3.1. Dataset and Implementation Details

We perform experiments on the publicly available CheXpert dataset (Irvin et al., 2019) that contains over 200,000 chest X-ray images, with binary labels for 14 diseases (e.g. Pleural Effusion, Cardiomegaly, Pneumonia), as well as binary labels indicating the presence of support devices (visual artifacts). We create two variants of the CheXpert dataset: (i) **Dot dataset**: We introduce a synthetic artifact, a black dot of radius 9 pixels, in the center of the image for evaluating the quality of counterfactual images in the presence of spurious correlations. This also helps to compare the behaviour of counterfactual images synthesized from the baseline method: DeCoDEx without a detector and the proposed DeCoDEx. (ii) **Device dataset:** A subset of the CheXpert data is used to demonstrate the performance of DeCoDEx in the presence of real artifacts (Support devices). In both datasets, the artifacts are present in the majority of images of subjects with Pleural Effusion and in the minority of images of healthy subjects. In contrast, the majority of the images of healthy subjects and the minority of the images of subjects with Pleural Effusion do not contain these artifacts. Fig. 3 shows the sample images from both the datasets and all four subgroups. For both datasets, the ratio of the number of samples in majority to minority is 90:10 and the dataset is divided into training/validation/testing with a 70/15/15 random split. The details of number of samples in different split is included in Appendix A.

The DenseNet-121 (Huang et al., 2017) architecture is used to train the classifier and detector. The classifier is trained separately on Dot and Device datasets. We use the standard Empirical Risk Minimization (ERM) (Sagawa et al., 2020) as the optimization

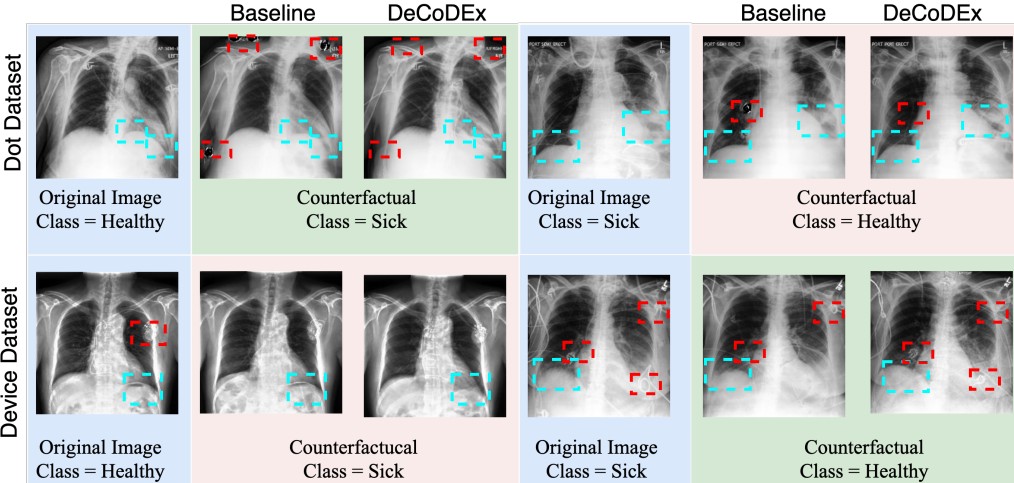

Figure 4: Qualitative comparison of counterfactual images synthesized via Baseline (i.e. DeCoDEx without detector) and DeCoDEx. For the baseline, most of the changes were made to the spurious correlation but for DeCoDEx visual artifacts were ignored and changes pertained to disease pathology.

method. The binary detector indicating the presence/absence of support devices is pretrained on the entire CheXpert dataset (except the held out test set). DeCoDEx is capable of handling images with multiple support devices, each varying in type, shape, size, location and intensities. An analysis showing the performance of the detector via counterfactual image generation is discussed in Appendix B.

### 3.2. Metrics and Experiments to evaluate counterfactuals

Several metrics are used to evaluate the quality of counterfactual images: (i) *Subject Identity Preservation: L1 Score* as depicted by the L1 distance between the counterfactual and the factual (original) image (as in (Mothilal et al., 2020; Nemirovsky et al., 2020)); (ii) *Counterfactual Prediction Gain (CPG)* (Nemirovsky et al., 2020) which measures the absolute value of the difference in the prediction of the classifier on the factual and counterfactual images (a higher score indicates a maximal change in the classifier decision boundary); (iii) *Spurious Correlation Latching Score (SCLS)*) (Kumar et al., 2023) assesses whether the spurious correlation was preserved in the counterfactual image (a lower SCLS score is desirable); (iv) *Classifier Flip Rate (CFR)* which represent the number of samples that flipped their class as per the classifier and (v) *Detector Robustness Rate (DRR)* showing number of samples that were robust to the detector.

In order to show that the synthesized counterfactuals learn useful features associated with the disease, the training data for the original classifier was augmented with synthesized images. The ERM classifier is retrained with the augmented data. An increase in performance indicates that synthesized images learned discriminative features generalizable to the

---

0. Only the training dataset is augmented while the validation and test split remains the same.

| | CFR↑ | DRR↑ | L1↓ | CPG↑ | SCLS↓ |
|---|---|---|---|---|---|
| | | | Dot dataset | | |
| Baseline | 0.8975 | 0.2575 | 0.038 | **0.592** | 0.7394 |
| DeCoDEx | **1** | **0.9775** | **0.036** | 0.559 | **0.058** |
| | | | Device dataset | | |
| Baseline | **0.97** | 0.7625 | 0.040 | 0.377 | 0.201 |
| DeCoDEx | 0.89 | **0.99** | **0.035** | **0.529** | **0.068** |

Table 1: Quantitative results comparing the scores for the counterfactuals generated by the Baseline and DeCoDEx on both datasets. Notice the high DRR and low SCLS values for DeCoDEx showing the spurious correlation were ignored in the counterfactual images.

subgroups. We augment the Dot dataset with 200 counterfactual images and the Device dataset with 600 counterfactual images synthesized using basline method and DeCoDEx. For completeness, the augmentation experiments are repeated for a debiasing Group-DRO classifier.

### 3.3. Results

**Classifier and Detector evaluation** The classifier's performance on both datasets is shown in Table 2 on the row labeled 'ERM'. Note that the performance on the minority subgroup samples is significantly lower than the majority for both datasets. The dot and device detectors perform very well with subgroups [$majority_S$, $minority_S$, $minority_H$, $majority_H$] accuracies of [100, 99.9, 100, 99.8] and [91.8, 88.8, 79.2, 88.7]. Perfect accuracies for the dot detector can be expected given that the position and size of the dot remain fixed for all the subjects. However, the variability including size, position, intensity of support devices can be large, making detection much more challenging.

**Qualitative evaluation** Pleural effusion (PE) is characterized by the rounding of the costophrenic angle, augmented lung opacity, and reduced clarity of the diaphragm and lung fissures (Light, 2002). It is observed in the lower corner of the lungs (Wang et al., 2017; Huang et al., 2022). Qualitative results for counterfactual generation from the models can be seen in Fig. 4. For the baseline method on the Dot dataset, the counterfactual for a healthy subject simply added dots to the image, indicating that the classifier has latched onto the dot artifact. However, DeCoDEx ignored the dot artifact (and maintained the original text artifact) and made changes reflective of PE disease. For the Device dataset, the baseline generated a counterfactual for a sick patient from the majority subgroup by simply removing the support device(s) while DeCoDEx makes (correct) changes in the area associated with the disease. Therefore, DeCoDEx indeed ignores the spurious correlation.

**Quantitative evaluation** In Table 1, CF images generated by the baseline and DeCoDEx show similar results for the metric L1 indicating CF images are similar to the factual image. CF synthesized by DeCoDEx have SCLS score close to zero indicating that the artifact was preserved in the counterfactual images. Table 2 shows the results of the augmentation experiments. Augmented training significantly improves the performance over the minority

| Dot dataset | | | | |
|---|---|---|---|---|
| | Pleural Effusion | | Healthy | |
| | Dot | No Dot | Dot | No Dot |
| ERM | 97 | 2.2 | 8 | 100 |
| ERM augmented with Baseline CFs | **98.5** | 0 | 16.6 | 100 |
| ERM augmented with DeCoDEx CFs | 90.6 | **12.1** | **53.1** | 98.0 |
| Group-DRO | 90 | 61.8 | 56.0 | 88.0 |
| Group-DRO augmented with Baseline CFs | **97** | 8.0 | 33.0 | **98.9** |
| Group-DRO augmented with DeCoDEx CFs | 91.0 | **70.2** | **60.9** | 80.3 |
| Device dataset | | | | |
| | Pleural Effusion | | Healthy | |
| | Support Device | No Support Device | Support Device | No Support Device |
| ERM | **92.7** | 75.2 | 84.9 | 87.0 |
| ERM augmented with Baseline CFs | 92.5 | 74.8 | 83.4 | 87.0 |
| ERM augmented with DeCoDEx CFs | 92.6 | **76.3** | **85.9** | 86.8 |
| Group-DRO | 92.7 | 83.4 | 77.3 | 88.4 |
| Group-DRO augmented with Baseline CFs | **97.8** | **85.4** | 55.9 | 72.0 |
| Group-DRO augmented with DeCoDEx CFs | 93.2 | 84.7 | **79.0** | 88.4 |

Table 2: Augmented Classifier Accuracies: CFs generated by DeCoDEx and the Baseline are used to augment the imbalanced datasets. Both ERM and Group-DRO are retrained on these augmented datasets and the effects are examined. The accuracies (percentages) of all classifiers are shown on the held out test set. Green indicates majority subgroups (90%) and red are the minority subgroups (10%). The best results are in bold. Note the increase in performance for the minority classes when both the methods, ERM and Group-DRO are augmented with De-CoDEx counterfactual images, illustrating the power of our method on extracting disease pathology and generating better counterfactual explanations.

groups for the Dot dataset, indicating that CF samples have learned discriminative features common to this subgroup. Among both ERM and Group-DRO based techniques, our method outperforms the Baseline along with augmented CFs. CF synthesized from DeCoDEx when augmented with ERM performs as well as Group-DRO for some minority classes which is a positive result, particularly when debiasing cannot be performed due to the lack of annotations.

## 4. Conclusions

In medical image analysis, explainable models are needed to expose and mitigate the bias to improve the trustworthiness of complex models. This paper presents DeCoDEx, an explainability framework that leverages a pre-trained classifier and detector to guide diffusion-based counterfactual synthesis towards accurate disease markers while ignoring spurious correlations. Qualitative and quantitative analysis of our extensive experiments indicate that the proposed method outperforms the baseline model that does not use a detector. Furthermore, the flexibility of our method allows it to be used with any pre-trained detector, does not require retraining a debiasing classifier and associated generative architecture, and provides guidance during inference. One of the current limitations of the model is resulting minor changes throughout the generated CF images. Future work will explore conditional score-based generative models.

## Acknowledgments

The authors are grateful for funding provided by the Natural Sciences and Engineering Research Council of Canada, the Canadian Institute for Advanced Research (CIFAR) Artificial Intelligence Chairs program, Mila - Quebec AI Institute, Google Research, Calcul Quebec, and the Digital Research Alliance of Canada.

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

## Appendix A. Detailed dataset description

We elaborate on the datasets and their variants used in our experiments in this appendix, complemented by statistical data on participant distribution across groups as detailed in Table 3.

| Dot dataset | | | | | | |
|---|---|---|---|---|---|---|
| **Spurious Correlation** | **Pleural Effusion** | | | **Healthy** | | |
| | Train | Validation | Test | Train | Validation | Test |
| Dot | 1359 | 191 | 242 | 56 | 5 | 49 |
| No Dot | 166 | 28 | 242 | 340 | 51 | 49 |
| **Device dataset** | | | | | | |
| **Spurious Correlation** | **Pleural Effusion** | | | **Healthy** | | |
| | Train | Validation | Test | Train | Validation | Test |
| Support Device(s) | 6653 | 1432 | 1389 | 665 | 143 | 138 |
| No Support Device(s) | 665 | 143 | 138 | 6653 | 1432 | 1389 |

Table 3:   Summary of the number of samples for both dataset variants

## Appendix B. Explainability: Providing insight into the detector result

We wish to provide some insights into the workings of the artifact/medical device detector. Fig. 5 shows two examples of explainability via counterfactual image generation illustrating the correct working of the detector. In both examples, the binary classifier was correctly focusing on the support devices. These are removed in the counterfactual images in order to flip the decision of the binary classifier.

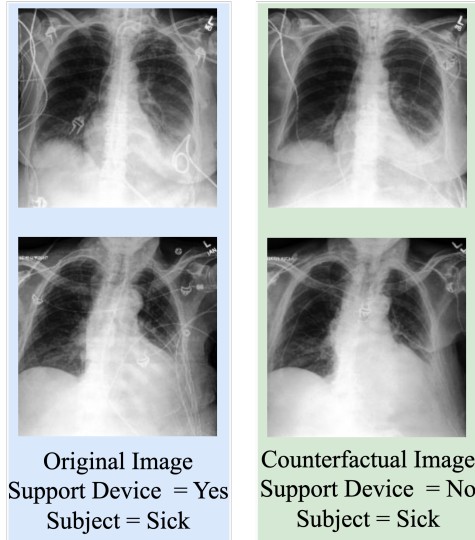

Figure 5: CF explanations for the detector: Removing medical devices from the original images while explaining the detector. Note the disease state is maintained in the counterfactual image.

## Appendix C. Extensive augmentation of the minority subgroup with synthesized CFs

| | Pleural Effusion | | Healthy | |
|---|---|---|---|---|
| *Dot dataset* | | | | |
| | Dot | No Dot | Dot | No Dot |
| ERM augmented with DeCoDEx CFs [200] | 90.6 | 12.1 | 53.1 | 98.0 |
| ERM augmented with DeCoDEx CFs [400] | **93.8** | **26.5** | **61.2** | 98.3 |
| Group-DRO augmented with DeCoDEx CFs [200] | 91.0 | 70.2 | 60.9 | 80.3 |
| Group-DRO augmented with DeCoDEx CFs [400] | **93.1** | **81.6** | **65.3** | 85.9 |
| *Device dataset* | | | | |
| | Support Device | No Support Device | Support Device | No Support Device |
| ERM augmented with DeCoDEx CFs [600] | 92.6 | 76.3 | 85.9 | 86.8 |
| ERM augmented with DeCoDEx CFs [1600] | **92.8** | **76.8** | **86.8** | 82.3 |
| Group-DRO augmented with DeCoDEx CFs [600] | 93.2 | 84.7 | 79.0 | 88.4 |
| Group-DRO augmented with DeCoDEx CFs [1600] | **93.5** | **86.5** | **79.9** | **90.1** |

Table 4: Improved accuracy at subgroup level through extensive classifier augmentation: Using DeCoDEx, we expanded on counterfactual generation to demonstrate improvement in the subgroup accuracy. We synthesise 400 counterfactual samples for the Dot dataset and 1600 for the Device dataset (the number in the square bracket refers to the total number of augmentation samples added to both majority and minority subgroups). First row is the original results discussed in Table 2 and the second row shows the result after extensive augmentation. Notice the improvement in the accuracy of minority subgroups across both Dot and Device datasets. Notably, 90% of these counterfactuals represent minority subgroups, thereby achieving a more equalized distribution in the dataset. Our findings indicate improvement in the accuracy of minority groups across all scenarios.

## Appendix D. Validating the Preservation of Patient Sex in the Synthesized Counterfactual Images

The identity preservation loss in Equation 3 does not guarantee that all the other attributes of the patients are maintained in the counterfactual images. A quick experiment was performed to validate that the sex of the patients is maintained in the counterfactual images generated by DeCoDEx. To this end, a sex classifier, $\mathcal{G}$, is trained on the real images, and then tested on real and synthesized counterfactual images. The sex classifier had an AUC-ROC of 0.98 on the real (factual) images. The differences in the sex classifier results based on the factual (F) and the counterfactual (CF) images, $|\mathcal{G}(\text{F}) - \mathcal{G}(\text{CF})|$, were 0.08 on average, indicating that the sex attribute was maintained in the counterfactual images.

