# OpenReview forum: "DeCoDEx: Confounder Detector Guidance for Improved Diffusion-based Counterfactual Explanations"
_MIDL.io/2024/Conference — MIDL 2024 Oral_

### Official Review · Reviewer_P5iS · 2024-02-27

**Confidence:** 4
**Preliminary Rating:** 4
**Final Rating:** 5

**Summary:**

This paper proposes DeCoDEx, a diffusion based framework to generate counterfactuals while ignoring visual artefacts. The framework involves a DDPM, a pre-trained classifier for disease, and a pre-trained detector for visual artefacts. The main idea is to generate counterfactuals with the guide of gradients from the classifier and detector to change/maintain disease area while ignoring artefacts.

**Strengths:**

-This paper is well-written and well-motivated.

-The authors have done extensive experiments to validate the method including several metrics and downstream task augmentation.

-In downstream augmentation, the results show that Group-DRO augmented with DeCoDEx CFs improved the worst-subgroup performance hugely, which is promising.

**Weaknesses:**

- The term “counterfactual” is overused in the field of image synthesis. Speaking in terms of causality theory, to produce a counterfactual, we need to define a causal graph and to abduct the background noises, such that all unrelated attributes remain the same. In this paper, the authors used identity preservation loss, however it may not guarantee that all unrelated attributes do not change. For instance, if we assume disease is not causally related to gender, then when we change disease, gender should not change as it is not affected. This is not discussed in the paper.
- The proposed method requires pre-trained disease classifier and pre-trained artefacts detectors, which are not always feasible. Have you consider using classifier-free guidance to get rid of reliance on pre-trained classifiers?
- The technical contribution/novelty of this paper is not clear compared to diffsuion works (e.g. Ho et al.). The benefit of "the flexibility of our method allows it to be used with any pre-trained detector” is actually from diffusion models. This paper is more like an application of classifier-guided diffusion sampling, not an improvement.

**Detailed Comments:**

- Table 1 and 2 could be put in a better place, rather than mixed with conclusion section.

**Justification Of Final Rating:**

This paper leverages DDPM, a pre-trained disease classifier and a pre-trained artefact detector to produce counterfactuals that ignore artifact. Although the technical novelty is limited compared to diffusion works, I think the application is clever and the problem it tries to solve is significant.

=====update======
The response addresses my concern about novelty. They also performed experiments to check if unintervened attribute remain the same.

**Justification Of The Preliminary Rating:**

This paper leverages DDPM, a pre-trained disease classifier and a pre-trained artefact detector to produce counterfactuals that ignore artifact. Although the technical novelty is limited compared to diffusion works, I think the application is clever and the problem it tries to solve is significant.

**Questions To Address In The Rebuttal:**

-In the paper, it is stated the previous diffusion works "with the classifier latching onto shortcuts present in the dataset”. How do the authors ensure the pre-trained disease classifier is not affected by shortcuts?
- See “Weakness".

---

> ### Author Response · Authors · 2024-03-16
>
> Thank you for your valuable and detailed comments. We are glad that you found our paper well-written and extensively researched.
>
> >..it is stated the previous diffusion works "with the classifier latching onto shortcuts present in the dataset”. How do the authors ensure the pre-trained disease classifier is not affected by shortcuts?
>
> We wish to reiterate that we do not claim the pre-trained disease classifier is not affected by shortcuts. In fact, it is quite the opposite. Our framework provides a counterfactual explanation method that ignores spurious correlations and, instead, explains the decision boundary of the classifier through the guidance of the detector.
>
> >The term “counterfactual” is overused in the field of image synthesis. Speaking in terms of causality theory, to produce a counterfactual, we need to define a causal graph and to abduct the background noises, such that all unrelated attributes remain the same. In this paper, the authors used identity preservation loss, however it may not guarantee that all unrelated attributes do not change. For instance, if we assume disease is not causally related to gender, then when we change disease, gender should not change as it is not affected. This is not discussed in the paper.
>
> We agree with the reviewer about the field of image synthesis, but addressing this point is beyond the scope of our current work. It is true that identity preservation loss does not guarantee that all the non-causal attributes remain the same. However, we do not have complete information about all the causal/ non-causal factors directly related to the disease under observation. We did run experiments whereby a classifier was built to identify the sex of the factual subject. We checked and indeed, the sex attribute remained stable for all real and counterfactual images in all cases. We can add these experiments to the Appendix if needed.
>
> >The proposed method requires pre-trained disease classifier and pre-trained artefacts detectors, which are not always feasible. Have you consider using classifier-free guidance to get rid of reliance on pre-trained classifiers?
>
> Using classifier-free guidance to address the same issue as presented in the paper could be an interesting direction for future research. However, classifier-free guidance does not address the issue of explainability, when the classifier latches onto spurious correlations. A pre-trained classifier has a number of other benefits in our field. In contexts where there is a relatively small dataset, training a robust detector would be challenging. Hence, a pre-trained detector trained on a different larger dataset can be used in our proposed framework in order to offset spurious correlations.
>
> >The technical contribution/novelty of this paper is not clear compared to diffsuion works (e.g. Ho et al.). The benefit of "the flexibility of our method allows it to be used with any pre-trained detector” is actually from diffusion models. This paper is more like an application of classifier-guided diffusion sampling, not an improvement.
>
> We do not claim that our proposed method presents the first classifier-guidance diffusion method nor that it presents an improvement over the original method in general. We show the benefit of _adapting_ a classifier-guidance diffusion strategy in real-world, important image analysis contexts. Our proposed method can overcome the effect of variable artifacts in medical images, which classifiers can latch onto when making decisions, thereby providing more informative and relevant (disease-related) counterfactual explanations.

---

> ### Comment · Reviewer_P5iS · 2024-03-20
>
> Thank the authors for the response. I update my rating accordingly. Congratulations on the great work.
>
> "We did run experiments whereby a classifier was built to identify the sex of the factual subject. We checked and indeed, the sex attribute remained stable for all real and counterfactual images in all cases. We can add these experiments to the Appendix if needed.”
>
> Yes, please. If space allows, why not.

---

> > ### Author Response · Authors · 2024-03-28
> >
> > Thank you. We have included the details in the _Appendix D_ of the manuscript.

---

### Official Review · Reviewer_LtCn · 2024-02-28

**Confidence:** 3
**Preliminary Rating:** 5
**Recommendation:** Oral

**Summary:**

The authors present a method to generate counterfactual images aiming to elucidate if a black-box classifier jumped on a confounder or the actual target. For this, they set up a training method for the generative model, and conduct extensive experiments to show the feasibility of their approach.

**Strengths:**

I liked the paper a lot. I was in particular happy about the educational, concise overview of the state of the art and the problems of similar approaches.

Most of the submission was easy to follow and structured logically. Choices seemed well motivated, and the presentation of results is mostly convincing. The conducted experiments are thoroughly documented, leading to a reasonable possibility to reproduce the publication. I hope some documentation in the already available code repo will detail how to do this.

**Weaknesses:**

I'm unfortunately late with my reviews, so I have to focus on the questions of greatest concern. They are few, anyways.
The main discussion point seems to be for me the artifact-specific classifier that needs to be provided for the method to work. This implies that I know of all artifacts that could confuse the method, and that I have a way of labeling the data accordingly. If there are multiple such artifacts, I wonder how I should proceed: Train one classifier? Or will I need to extend the method to have one loss per detected artifact? Or move from a binary to a n-ary classifier and a more complicated loss? I'd like to raise this topic for the rebuttal.
Another point was the role of the classifier and the detector in the generation pipeline. Is it correct that the classifier is the "given" entity for which the explanation should be achieved, by using the artifact detector to generate the counterfactual image -- and the wrong classifier decision makes the detector kick in to drive the generative process? I think the thought model behind this setup could use a few more words.

**Detailed Comments:**

Sorry to omit this due to the brevity of time at my disposal. There wasn't a lot, though.

Looking at the first figure with clinical images, I wished for a comparative presentation of true positives/negatives with the expected appearance indicated. This is accomplished in a later figure. Perhaps it makes sense to pull parts up.

You speak of "established metrics", citing a publication that is hardly 1 year old. Perhaps the wording "established" isn't quite what you should use in this case.

Calling the detector a "detector" is also slightly misleading -- isn't it also "just" a classifier indicating the global presence or absence of the artifact?

**Justification Of The Preliminary Rating:**

I think this submission inspires. Besides its clear presentation, it deals with a subject of great importance in the medical field, has potential for application in other medical domains and not least provides a code basis (which is not yet documented, though).

**Questions To Address In The Rebuttal:**

See above.

---

> ### Author Response · Authors · 2024-03-16
>
> Thank you for the detailed review and positive feedback.
>
> Thank you for your minor suggestions which we implemented. We moved the example figure to the front of the paper. We omitted the word ‘established’ for a recently published metric. We made our entire codebase public and documented the complete installation process.
>
> With reference to your question, while the detector is indeed a classifier (indicating the global presence or absence of artifacts), we named it a “detector” in order to avoid confusion with the disease “classifier” in the model. We also wish to clarify that our model is capable of handling different artifacts in factual images. We clarified this in the paper to make sure that this is evident (Section 3.1, page 6):
>
> “The detector is trained with binary target labels indicating the presence/absence of support devices.  DeCoDEx is capable of handling images with multiple artifacts. These can vary in type, shape, size, location and intensities.”

---

### Official Review · Reviewer_R6Gn · 2024-03-01

**Confidence:** 4
**Preliminary Rating:** 4

**Summary:**

This paper presents a novel framework aimed at improving the explainability of deep learning classifiers in medical imaging. It specifically addresses the issue where classifiers rely on spurious, non-causal features (confounders) in datasets, leading to biased predictions. The proposed DeCoDEx framework leverages a pre-trained binary artifact detector during inference to guide the synthesis of counterfactual images by a diffusion-based generator, focusing on causal features and ignoring irrelevant artifacts. The methodology involves generating counterfactuals while preserving subject identity and ensuring classifier consistency, using a denoising diffusion probabilistic model (DDPM) guided by the artifact detector. Extensive experiments on the CheXpert dataset, using both synthetic and real artifacts, demonstrate that DeCoDEx improves the quality of counterfactual explanations and classifier performance, particularly for underrepresented groups. The paper suggests that integrating DeCoDEx with existing classifiers can enhance bias mitigation and model trustworthiness without requiring retraining or subgroup labels.

**Strengths:**

This is a successful application of diffusion model to medical imaging. There are convincing results presented in this paper, showing that the proposed DeCoDEx model out-performs other published baselines on benchmarking datasets. In addition, the analysis in this paper is well-conducted, described the cause of confounders in medical images clearly. The paper is well organized and well-written. No English grammar error detected in this paper.

**Weaknesses:**

The paper identifies a limitation within its proposed DeCoDEx framework, namely that it results in minor changes throughout the generated counterfactual (CF) images. This could potentially limit the interpretability and granularity of the explanations provided by the model. Additionally, the paper mentions that future work will explore conditional score-based generative models, which implies that the current model may not fully capture or manipulate the complex dependencies or conditions present in medical imaging data​. Finally, there is no ablation study in this paper. I suggest that the authors provide an ablation study to further discover which part of the model contributes most to the performance. In this way, people can understand the proposed DeCoDEx better intuitively.

**Detailed Comments:**

This paper presents a novel framework aimed at improving the explainability of deep learning classifiers in medical imaging. It specifically addresses the issue where classifiers rely on spurious, non-causal features (confounders) in datasets, leading to biased predictions. The proposed DeCoDEx framework leverages a pre-trained binary artifact detector during inference to guide the synthesis of counterfactual images by a diffusion-based generator, focusing on causal features and ignoring irrelevant artifacts. The methodology involves generating counterfactuals while preserving subject identity and ensuring classifier consistency, using a denoising diffusion probabilistic model (DDPM) guided by the artifact detector. Extensive experiments on the CheXpert dataset, using both synthetic and real artifacts, demonstrate that DeCoDEx improves the quality of counterfactual explanations and classifier performance, particularly for underrepresented groups. The paper suggests that integrating DeCoDEx with existing classifiers can enhance bias mitigation and model trustworthiness without requiring retraining or subgroup labels.

This is a successful application of diffusion model to medical imaging. There are convincing results presented in this paper, showing that the proposed DeCoDEx model out-performs other published baselines on benchmarking datasets. In addition, the analysis in this paper is well-conducted, described the cause of confounders in medical images clearly. The paper is well organized and well-written. No English grammar error detected in this paper.

The paper identifies a limitation within its proposed DeCoDEx framework, namely that it results in minor changes throughout the generated counterfactual (CF) images. This could potentially limit the interpretability and granularity of the explanations provided by the model. Additionally, the paper mentions that future work will explore conditional score-based generative models, which implies that the current model may not fully capture or manipulate the complex dependencies or conditions present in medical imaging data​. Finally, there is no ablation study in this paper. I suggest that the authors provide an ablation study to further discover which part of the model contributes most to the performance. In this way, people can understand the proposed DeCoDEx better intuitively.

In general, this is a successful application of deep neural networks (diffusion models) to medical imaging. There are novelty proposed in the model structure, and the model out-performs other recently published baselines. Although there are detects in the model framework, they cannot be the reason to deny the advantage of the proposed model. So, I recommend to accept this paper after minor revision.

**Justification Of The Preliminary Rating:**

In general, this is a successful application of deep neural networks (diffusion models) to medical imaging. There are novelty proposed in the model structure, and the model out-performs other recently published baselines. Although there are detects in the model framework, they cannot be the reason to deny the advantage of the proposed model. So, I recommend to accept this paper after minor revision.

**Questions To Address In The Rebuttal:**

Not applicable

---

> ### Author Response · Authors · 2024-03-16
>
> Thank you for your valuable feedback.
>
> Based on the reviewer's suggestions, we added a set of additional ablations to the paper related to the number of synthesized samples we used for augmentation (Appendix C). The purpose of the experiments was to examine whether additional augmentations provided to the minority subgroups would improve their classification accuracies. In the original paper, we performed augmentation experiments where we illustrated that the addition of 200 synthesized counterfactual images improved the classification performance for the minority subgroups. We now perform experiments where we increased the augmentation by 400 synthesized images for the dot dataset and 1600 synthesized images for the device dataset. A greater augmentation of the original dataset (90% minority and 10% majority) showed significant improvements in the performance of the minority subgroups. We show the additional table (Appendix C) here:
>
> **Dot Dataset**
> |                                        |        **Pleural Effusion**         |                             |       **Healthy**        |                      |
> |--------------------------------------------|:-----------------------------------:|:---------------------------:|:------------------------:|:--------------------:|
> |                                        |              **Dot**                |          **No Dot**         |          **Dot**         |      **No Dot**      |
> ||||||
> | **ERM augmented with DeCoDEx CFs [200]**      |              90.6                   |            12.1             |          53.1            |          98.0        |
> | **ERM augmented with DeCoDEx CFs [400]**      |       **93.8**      |       **26.5**          |       **61.2**          |      **98.3**      |
> ||||||
> | **Group-DRO augmented with DeCoDEx CFs [200]** |              91.0                   |            70.2             |          60.9            |          80.3        |
> | **Group-DRO augmented with DeCoDEx CFs [400]** |       **93.1**     |        **81.6**          |       **65.3**          |      **85.9**      |
>
>
> **Device Dataset**
> |                                         |       **Pleural Effusion**        |                            |       **Healthy**        |                    |
> |-------------------------------------------------|:--------------------------------------:|:-------------------------:|:----------------:|:------------------------:|
> |                                         |         **Support Device**        |    **No Support Device**   |    **Support Device**    | **No Support Device** |
> ||||||
> | **ERM augmented with DeCoDEx CFs [600]**      |              92.6                  |            76.3            |          85.9            |          86.8      |
> | **ERM augmented with DeCoDEx CFs [1600]**     |       **92.8**     |        **76.8**         |       **86.8**          |      82.3      |
> ||||||
> | **Group-DRO augmented with DeCoDEx CFs [600]** |              93.2                  |            84.7            |          79.0            |          88.4      |
> | **Group-DRO augmented with DeCoDEx CFs [1600]**|       **93.5**     |        **86.5**         |       **79.9**          |      **90.1**    |

---

### Author Response · Authors · 2024-03-16

We would like to thank all the reviewers for their valuable insights and thoughtful review of our paper. We appreciate the thoroughness with which each reviewer reviewed our manuscript, offering insightful suggestions and highlighting areas of improvement.

We are happy that the reviewers found our paper well-written and well-organized.  We are encouraged with the general consensus that this work presents a successful adaptation of diffusion models in the context of addressing biases in classifier predictions. We have made our code public, for ease of implementation and reproducibility of our approach. We documented our code and it should now be easy to understand and to use.

We have responded to each reviewer’s comments individually below and would be happy to address any comments or concerns.  We addressed each of the concerns and made the associated changes in the original manuscript. This includes the addition of ablation experiments, as requested, which we placed in the appendix. Detailed changes are described below.  We thank the reviewers for their suggestions as we feel they helped improve the paper.

---

### Meta-Review · Area_Chair_FLNN · 2024-04-04

**Recommendation:** Accept (Poster)
**Confidence:** 4

**Metareview:**

One reviewer did not update their recommendation and, from my appreciation of their concerns and the response, I am assuming they would be only partially satisfied as the response addresses the issue pertaining to ablations but does not address the potential weaknesses of minor changes of the counterfactuals and complex dependencies. Considering the overall positivity of this review, however, I will assume that this shortcoming would not result in a dramatically lower verdict. Thus, after revision, there is reviewer consensus that the manuscript can be accepted. The additional experiments should be included in the final manuscript.

---

### Decision · Program_Chairs · 2024-04-06

Accept (Oral)